# Intraoperative Hemoadsorption (Cytosorb™) during Open Thoracoabdominal Aortic Repair: A Pilot Randomized Controlled Trial

**DOI:** 10.3390/jcm12020546

**Published:** 2023-01-09

**Authors:** Panagiotis Doukas, Gabriel Hellfritsch, Daniel Wendt, Mirko Magliani, Mohammad E. Barbati, Houman Jalaie, Michael J. Jacobs, Alexander Gombert

**Affiliations:** 1European Vascular Centre Aachen-Maastricht, Department of Vascular Surgery, RWTH Aachen, 52074 Aachen, Germany; 2Westgerman Heart & Vascular Center, University Hospital Essen, 45147 Essen, Germany; 3CytoSorbents Inc., Princeton, NJ 08540, USA; 4Department of Cardiothoracic Surgery, RWTH Aachen, 52074 Aachen, Germany

**Keywords:** aortic surgery, hemoadsorption, cardiopulmonary bypass, inflammation

## Abstract

Background: The efficacy of cytokine adsorption in controlling the early inflammation cascade after open thoracoabdominal aortic (TAAA) repair has not been investigated. The aim of this pilot randomized controlled trial was to assess the feasibility and effect of perioperative hemoadsorption during open TAAA repair. Methods: Patients scheduled for open TAAA repair with the use of cardiopulmonary bypass (CPB) were included. The patients were randomized the day before surgery to either intraoperative hemoadsorption during CPB or standard of care. Results: A total of 10 patients were randomly assigned to the intervention group, whereas the control group consisted of 17 patients (mean age of the total cohort, 51.1 ± 11.2 years, 67% male, 3 patients not randomized). The majority of baseline and perioperative characteristics were similar, and no device-related adverse events were reported. A trend to shorter ventilation times in the intervention group was observed (median 88 h vs. 510 h, *p* = 0.08, Δ422). Severe acute respiratory distress syndrome was significantly less in the intervention patients (*p* = 0.02). Conclusions: This is the first pilot study showing that the intraoperative use of hemoadsorption in open TAAA repair patients may be feasible and safe, yet larger trials are needed to evaluate whether intraoperative hemoadsorption is associated with improved clinical outcomes.

## 1. Introduction

The main trend of treatment modalities for thoracoabdominal aortic aneurysms (TAAA) has shifted towards endovascular procedures in recent years [1]. However, open aortic repair remains the only curative treatment option for some patients and, despite optimized surgical protocols and meticulous postoperative management, it is still associated with a significant risk for early postoperative complications [2]. Indeed, data from high-volume centers report a mortality rate of 6.2% for elective cases, which may rise to 12.2% for urgent repairs [3]. Among other factors such as ischemic events and infections, the invasive nature of open surgery poses a relevant hindrance to the patient’s recovery.

Tissue injury related to thoraco-laparotomy to approach the aorta as well as to the use of extracorporeal circulation induces a systemic inflammatory response [4,5]. This effect is further aggravated by the ischemia-reperfusion injury of the viscera following aortic cross clamping [6,7]. The non-specific activation of cytokines and inflammation mediators, including Tumor Necrosis Factor alpha (TNF-α) and interleukins—IL1β, IL4, IL6, IL8 and IL10—may compromise the ability of the immune system to resist infection [8], dysregulates coagulation [9] and is associated with pulmonary organ dysfunction [10]. In the worst scenario, this systemic inflammatory response may be orchestrated by the high levels of circulating cytokines, leading to the phenomenon termed “cytokine storm” [11].

Evidently, reducing the levels of circulating pro-inflammation mediators through extracorporeal blood purification using hemoadsorption represents one possibility for cytokine removal with the use of sorbents [12]. Such devices are filled with biocompatible polymer sorbent beads that can remove hydrophobic substances, including cytokines. It has been shown that CytoSorb^®^ (CytoSorbents, Princeton, NJ, USA) promotes hemodynamic stability in critically ill patients with sepsis-associated acute kidney injury (KDIGO III) after cardiac surgery [13]. The intraoperative application of the system in the context of infective endocarditis reduced the cytokine load postoperatively in a multicenter, randomized controlled trial but did not affect postoperative organ dysfunction [14]. The potential of intraoperative hemoadsorption for controlling the early inflammation cascade after open thoracoabdominal aortic repair has not yet been investigated.

The aim of this pilot randomized controlled trial was therefore to assess the potential protective effect of perioperative cytokine adsorption during open TAAA repair to prevent complications through the early postoperative phase.

## 2. Materials and Methods

### 2.1. Study Design

The present study was a prospective, randomized, non-blinded, parallel, controlled pilot trial. The study was reviewed and authorized by the local ethics committee of the University Hospital Aachen (EK 004/14) and was performed in accordance with the Declaration of Helsinki and the CONSORT (Consolidated Standards of Reporting Trials) criteria. The study was registered on clinicaltrials.gov under the number NCT04765748. Written informed consent was obtained from all included patients.

All patients planned for open TAAA—urgent or elective cases—were considered for recruitment. Exclusion criteria were pregnancy and age below 18 years. The day before surgery, the patients were randomly assigned to the intervention group (*n* = 10) or the control group (*n* = 17, of which the first 3 were not randomized and served as an initial experience for a run-in phase but were then included in the final analysis). The sealed envelope method was used for randomization: the envelopes were created from personnel not involved in this research and were opened from the recruiting researchers the day before surgery.

Medical history and laboratory parameters were assessed from the digitally maintained patients’ charts (IntelliSpace Critical Care and Anesthesia; Philips Healthcare, Andover, MA, USA). Blood samples were collected before surgery, after admission to the intensive care unit (ICU) and during ICU stay (12, 24, 48, and 72 h).

### 2.2. Surgery

The surgical protocol for open TAAA repair was followed as described in previous publications [15,16] and included aortic cross-clamping, extracorporeal circulation with distal aortic perfusion via femoro-femoral cannulation and visceral perfusion using selective perfusion catheters. All procedures were performed under mild hypothermia (down to 35 °C). All patients were intubated with a double-lumen tube, which allowed for one-lung ventilation after opening the left thoracic cavity to approach the descending aorta. Both lungs were ventilated at all times when the descending aorta did not need to be exposed. The renal arteries were selective perfused with 1000 mL of Custodiol^®^ crystalloid solution (Dr. Franz Köhler Chemie, Bensheim, Germany) during the repair. In the intervention group the CytoSorb^®^ adsorber, a single-use, polymer bead-based cartridge, was integrated into the extracorporeal circuit. In the control group, no adsorber system was used.

### 2.3. Device

The CytoSorb^®^ 300 mL device (CytoSorbents, Princeton, NJ, USA) is filled with highly biocompatible, porous polymer beads covered with a divinylbenzene coating. Each polymer bead is between 300 and 800 μm in size and has pores and channels resulting in an effective surface area of more than 40,000 m^2^, capable of binding hydrophobic small- and medium-sized molecules [6]. Cytosorb^®^ is CE-marked according to the Medical Devices Directive (ISO 10993 biocompatible, manufactured in the United States under ISO 13,485 certification). The device and the intraoperative setup are displayed in Figure 1.

### 2.4. Endpoint Definition

To investigate the impact of cytokine adsorption during open TAAA surgery, we examined and compared the dynamics of inflammation markers, catecholamine requirements and clinical parameters between the two groups. Our hypothesis was that the application of the CytoSorb^®^ adsorber could attenuate inflammation in the early postoperative phase and contributed to hemodynamic and respiratory stability, which translated clinically to lower catecholamine requirements and fewer hours of invasive ventilation. The degree of required hemodynamic support through inotropes and vasopressors to maintain a mean arterial pressure of 65 mmHg was quantified with the vasoactive-inotropic score (VISmax) and was calculated according to the formula VISmax = dopamine dose (µg/kg/min) + dobutamine dose (µg/kg/min) + 100 × epinephrine dose (µg/kg/min) + 10 × milrinone dose (µg/kg/min) + 10,000 × vasopressin dose (units/kg/min) + 100 × norepinephrine dose (µg/kg/min) [17].

The secondary endpoints assessed mid- and long-term outcomes: sepsis, death, infections, neurological complications, re-interventions and MACE (major adverse cardiovascular events). Renal failure was defined according to the KDIGO (Kidney Disease: Improving Global Outcomes) classification [18]. Sepsis was defined as a life-threatening organ dysfunction as a result of the dysregulated host response to an infection according to “The Third International Consensus Definitions for Sepsis and Septic Shock” [19]. Neurological complications included the newly onset of paraplegia or paraparesis as a result of spinal cord ischemia [16]. Acute myocardial infarction, stroke and cardiovascular mortality were summarized as three-point MACE [20]. Acute respiratory distress syndrome (ARDS) was classified as mild, moderate or severe according to the Berlin criteria [21]. Prolonged ventilation was defined as greater than 21 days of mechanical ventilation for at least 6 h per day [22].

### 2.5. Statistics

The data were analyzed using GraphPad Prism version 9.0 software (GraphPad Software, San Diego, CA, USA). The continuous variables were expressed as mean ± standard deviation (SD) or as median [Interquartile range] in case of heavy skewness of the data. The Student’s t-test or the Mann–Whitney test were used to examine statistical significance. Categorical data were expressed as number of patients and frequencies and compared with the chi-square test and the Fisher’s exact test. *p*-values < 0.05 were considered significant.

## 3. Results

### 3.1. Baseline Characteristics

Between October 2019 and February 2022, a total of 27 patients were included into the present analysis. A total of 10 patients were randomly assigned to the intervention group, whereas the control group consisted of 17 patients (mean age of the total cohort 51.1 ± 11.2 years, 67% male; 63% were reoperation cases, 3 patients were not randomized). The two groups were comparable regarding their baseline characteristics. Moreover, also regarding their underlying diseases and aortic pathologies, no significant differences between the groups were detected. The preoperative baseline characteristics of the patients are displayed in detail in Table 1.

### 3.2. Operative and Postoperative Outcomes

Most perioperative factors were comparable between the groups. The mean CPB duration in the intervention group was 149 ± 46 min and did not differ with respect to the control group (*p* = 0.9). No device-related adverse events were reported. There were no differences between the groups for the extent of aortic involvement (Crawford classification). The majority of patients presented with Crawford 2, 3 and 4 classifications. Mild hypothermia and the thoraco-laparotomy approach were applied in all cases.

Major adverse cardiac events only occurred in the control group without being statistically significantly different and was driven by neurological events only. Mortality also did not differ between the two groups. Regarding the postoperative outcomes, postoperative dialysis did not differ between the groups; however, it was present in 50% of the intervention patients and in 70.1% of the control patients. We could not detect a significant difference in the maximum postoperative vasoactive-inotropic score for the whole ICU stay (32.1 ± 35.8 vs. 35.9 ± 45.5, *p* = 0.58) or for the first 24 h after surgery (42.8 ± 64 vs. 32.12 ± 35.8, *p* = 0.64). However, a non-significant trend to shorter ventilation times in the intervention group could be observed (median 88 h vs. 510 h, *p* = 0.08, Δ422), and the intervention group required significantly less often prolonged ventilation (1 vs. 9, *p* = 0.03). Moreover, severe ARDS was observed only in the control patients (0 vs. 41.1%, *p* = 0.02).

Figure 2 exemplifies the postoperative course of the CRP levels. The intervention group showed higher pre-operative CRP levels compared to the control group, but the difference was not statistically significant. The increase in CRP on postoperative day 1, however, could be attenuated by intraoperative hemoadsorption (increase in pre- vs. postoperative CRP in the intervention group, ΔCRP: 6.8 ± 100.3 vs. 73.0 ± 89.5 mg/dL, *p* = 0.08) without reaching significance.

The operative characteristics and post-operative outcome results are presented in Table 2. 

## 4. Discussion

The present pilot randomized-controlled trial investigated for the first time the potential protective effect of intraoperative cytokine adsorption during open TAAA repairs to prevent complications through the early postoperative phase. The main observations from the current study are: intraoperative hemoadsorption in TAAA repair patients is feasible and safe despite the high-risk nature of TAAA repair; the increase of CRP (although not being significant) could be attenuated despite the higher preoperative CRP levels in the hemoadsorption group; moreover, a non-significant trend to shorter ventilation times in the intervention group could be observed. Interestingly, severe ARDS only occurred in the control patients (statistically significant). However, we would like to clearly state that our findings apply only to patients treated with an open surgical reconstruction of the thoracoabdominal aorta.

Open thoracoabdominal surgery is a physiologically demanding procedure [23,24], with postoperative organ failure being a relevant and potentially fatal complication [25]. One contributing factor to this event is the non-specific systemic hyperinflammation mediated by a cytokine surge during and after surgery [26]. The physiological aftermath of the cytokine storm is hemodynamic instability and susceptibility to infections, which pose a risk for the development of sepsis and multiple organ dysfunction syndrome (MODS) [8]. In many cases, the centerpiece of MODS is the respiratory tract, often in the absence of a primary pulmonary focus [8]. Sepsis-induced lung injury translates on a cellular level to a disturbance of the alveo-capillary unit integrity. On top of the mechanical consequences of diaphragmatic incision and lateral thoracotomy to access the thoracic aorta, alveolar dysfunction is an additional obstacle to the patient’s early weaning from invasive ventilation [27].

To control the dysregulated immune homeostasis, blood purification techniques aim to eliminate the pro-inflammatory mediators from the circulation. One of these techniques, i.e., hemoadsorption with the CytoSorb column, adsorbs mid-molecular-weight solutes and binds them according to their intrinsic hydrophobic interactions, effectively removing them from the bloodstream [28]. As an adjuvant to renal replacement therapy for patients in septic shock, the use of the CytoSorb column is associated with decreased observed vs. expected mortality rates [29,30], improvement in hemodynamic stability [31] and shorter length of stay in the ICU [32]. Integrating the CytoSorb adsorber into the extracorporeal circulation assembly during cardiac surgery serves as the theoretical premise that cytokines induced through surgical trauma and the very application of cardiopulmonary bypass are eliminated at the time of their synthesis, thus promoting hemodynamic stability in the early postoperative phase, which was proven by the REMOVE trial [14]. In the context of infective endocarditis, hemoadsorption with CytoSorb resulted in less dependency on catecholamine therapy for patients undergoing dialysis postoperatively [13,33,34]. As mentioned above, the results of the multicenter, randomized, controlled REMOVE study (Cytokine Hemoadsorption During Cardiac Surgery Versus Standard Surgical Care for Infective Endocarditis) were published [14]. The primary outcome was the difference in change in the sequential organ failure assessment (SOFA) score. The REMOVE trial failed to show a reduction in the SOFA score in the postoperative course with intraoperative hemoadsorption; however, a significant reduction in cytokines was observed [14]. One should consider that the effect of hemoadsorption depends on the length of its application, which was also shown in the REMOVE trial: longer CPB times amplified the amount of IL-6 generation [14]. Incorporating the hemadsorption cartridge into continuous renal replacement therapy implies a longer period of cytokine removal compared to just its intraoperative use during CPB. Thoracoabdominal aortic (TAAA) repair, due to its high-risk nature, the excessive intraoperative trauma and potential long CPB times, seems to be the ideal intervention to evaluate adjunctive hemoadsorption techniques. In the case of open TAAA repair, the surgical trauma may be considered more extensive than in cardiac surgery [4], and the provoked ischemia-reperfusion injury of the viscera [6] may result in elevated plasma IL-6 and IL-8 concentrations, despite the selective perfusion of the intestinal arteries. Accordingly, the patients undergoing extensive aortic reconstructions are prone to elevated inflammatory activity in the early postoperative phase. In this way, investigating the effects of hemoadsorption during elective TAAA repair addresses a different setting of cytokine induction than the REMOVE trial, which did not prove a clinical benefit for intraoperative hemoadsorption regarding organ function after cardiac surgery for infective endocarditis.

Our hypothesis that the application of the CytoSorb^®^ adsorber could attenuate inflammation in the early postoperative phase and contributes to hemodynamic and respiratory stability could, in part, be proven. Although we did not observe a statistically significant difference in the maximum vasoactive-inotropic score, intraoperative hemoadsorption translated to fewer hours of invasive ventilation. As described above, pulmonary dysfunction is common in sepsis-associated MODS, even without primary lung infection. Albeit being non-significant, the median for invasive ventilation in the intervention group was 422 h shorter, and these patients required significantly less often prolonged mechanical ventilation. This finding is reinforced by the fact that respiratory failure occurred significantly less often in the hemoadsorption group than in the control group, with no severe ARDS being present in the hemoadsorption group. Acute respiratory distress syndrome is observed in around 44% of patients in septic shock [35]. One could speculate that controlling the initial inflammatory response addresses the issue of lung injury at an early stage and may efficiently limit the incidence of severe ARDS, enabling an expedited weaning from the ventilator.

Finally, the increase in CRP on postoperative day 1 was attenuated by intraoperative hemoadsorption. Although this finding was not significant, the overall difference between pre- and postoperative CRP was less in the hemoadsorption group, even though the intervention group started at a higher preoperative CRP level.

Overall, in-hospital mortality was high and comparable between the two groups. Among the in-hospital deaths, all patients died due to multiorgan failure, and all developed liver failure. In addition, these patients showed the highest vasoactive-inotropic score of the whole cohort. Moreover, a substantial percentage of patients developed postoperative severe kidney injury (KDIGO 3) requiring renal support. In the present analysis, 50% of the intervention group and 70% of the control group received postoperative continuous renal replacement therapy. This offers the potential of the postoperative continuation of hemoadsorption, which was not part of the protocol for this pilot trial.

## 5. Perspectives

Overall, the present study underlines the need for future multicenter, randomized, control trials, pooling all cases of open TAAA repairs, a relatively infrequently performed procedure. In this perspective, placing the focus of future trials on the maximum VIS in the first postoperative days (reported as VIS max 24 h in this cohort, with a mean value of 42.8 ± 64) may clarify the impact of intraoperative hemoadsorption in the patient’s hemodynamic stability. Lastly, investigating the incidence of ARDS and prolonged ventilation in a study with adequate statistical power could answer the question of the clinical benefit for the intervention group definitively.

## 6. Limitations

The outcomes of the present pilot controlled, randomized trial were primarily limited by the small cohort investigated. Moreover, we included three patients in the “roll-in phase” into the control group before randomization. On the other hand, the cases included in this single-center study were all operated by the same surgeon, eliminating the performance bias. Furthermore, our group mainly focused on clinical data and routinely acquired parameters. Although the lack of assessment of the cytokine plasma levels at baseline and postoperatively limits this study, it remains an intriguing topic for future research. Whether postoperative therapy continuation might have a further beneficial impact on clinical outcomes was not evaluated in the present analysis, despite the fact that the majority of the patients developed acute kidney injury in the postoperative period.

## Figures and Tables

**Figure 1 jcm-12-00546-f001:**
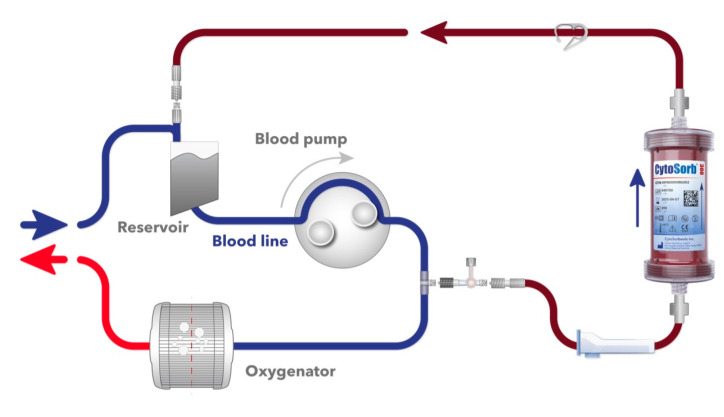
Illustration of the CytoSorb adsorber incorporated into the CPB circuit.

**Figure 2 jcm-12-00546-f002:**
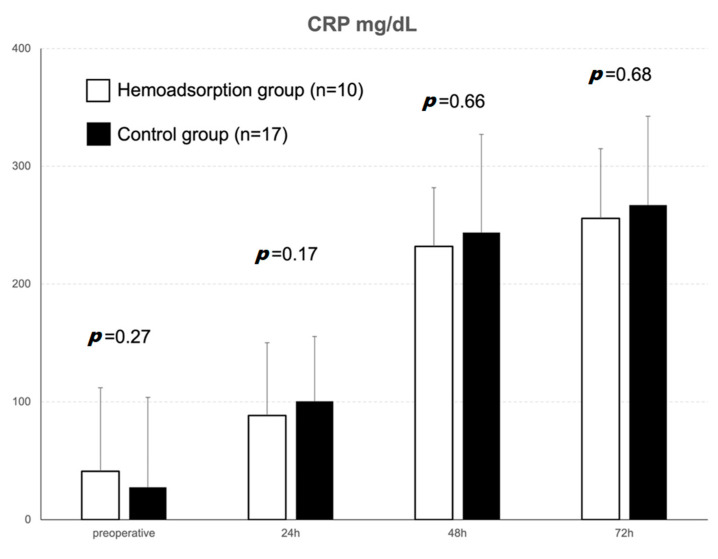
Course of CRP levels up to 72 h postoperatively (mg/dL).

**Table 1 jcm-12-00546-t001:** Baseline characteristics.

Variable	HA-Group *n* = 10	Control Group *n* = 17	*p*
**Demographics**			
Age, years	49.7 ± 13.1	50.5 ± 11.1	0.71
Gender, male	7 (70)	11 (64.7)	1
BMI, m^2^	23.5 ± 5.4	24.4 ± 4.3	0.69
COPD	2 (20)	3 (17.6)	1
Systemic hypertension	8 (80)	13 (76.4)	1
Smoking	4 (40)	4 (23.5)	0.64
Prior stroke	1 (10)	3 (17.6)	1
Diabetes	2 (20)	0 (0)	0.12
Atrial fibrillation	2 (20)	0 (0)	0.12
Emergency	2 (20)	1 (5.8)	0.53
Renal insufficiency	6 (60)	8 (47.1)	0.69
Reoperation	6 (60)	11 (64.7)	1
Cardiac re-operation	1 (10)	5 (29.4)	0.36
Heart failure (EF < 30%)	2 (20)	3 (29.4)	1
Aortic aneurysm diameter	5.8 ± 1.6	6.1 ± 1.2	0.68
**Aortic pathology**			
True aneurysm	1 (10)	4 (23.5)	0.62
Mycotic including rupture	2 (20)	2 (11.8)	0.61
Post-dissection	4 (40)	9 (52.9)	0.69
Dissection	1 (10)	0 (0)	0.37
Marfan syndrome	1 (10)	2 (11.8)	1
Anastomotic aneurysm	1 (10)	0 (0)	0.37

Continuous variables are presented as mean ± SD, and categorical variables are presented as absolute frequencies (%); BMI: body mass index; COPD: chronic obstructive pulmonary disease; EF: ejection fraction.

**Table 2 jcm-12-00546-t002:** Operative and postoperative characteristics.

Variable	HA-Group *n* = 10	Control Group *n* = 17	*p*
**Operative characteristics**			
CPB, min.	149 ± 46	152 ± 46	0.90
Total OR time, min.	489 ± 125	502 ± 89	0.92
Y-prosthesis	2 (20)	5 (29.4)	0.67
Mild hypothermia (down to 35 °C)	10 (100)	17 100)	1
Thoraco-laparotomy approach	10 (100)	17 (100)	1
**Crawford classification**			
Type 1	0 (0)	1 (5.8)	1
Type 2	5 (50)	6 (35.3)	0.68
Type 3	3 (30)	5 (29.4)	1
Type 4	1 (10)	4 (23.5)	0.62
Type 5	1 (10)	0 (0)	0.37
**Blood products**			
Packed red blood cells, bags	22.5 ± 18.3	19.8 ± 11.1	0.63
Fresh-frozen plasma, bags	30.3 ± 19.8	23.4 ± 10.8	0.25
Platelet concentrate, bags	6.6 ± 3.7	4.6 ± 1.8	0.07
**Postoperative outcomes**			
Rethoracotomy	4 (40)	9 (52.3)	0.69
Dialysis	5 (50)	12 (70.1)	0.41
Sepsis	3 (30)	7 (41.1)	0.69
Ventilation time, h	88 [9–337]	510 [14–954.5]	0.08
Prolonged ventilation	1 (10)	9 (52.9)	0.03 *
Tracheostomy	4 (40)	11 (64.7)	0.23
Pneumonia	7 (70)	11 (64.7)	0.79
ARDS (severe)	0 (0)	7 (41.1)	0.02 *
New stroke	0 (0)	5 (29.4)	0.12
Myocardial infarction	0 (0)	0 (0)	1
Cardiac-related death	0 (0)	0 (0)	1
MACE (composite)	0 (0)	5 (29.4)	0.12
Septic shock	1 (10)	4 (23.5)	0.62
Hospital stay, days	46.4 ± 30.1	49.7 ± 34.6	0.51
VISmax	35.9 ± 45.5	32.1 ± 35.8	0.58
VISmax 24 h	42.8 ± 64	32.12 ± 35.8	0.64
Liver failure	3 (30)	2 (11.8)	0.32
30-day mortality	2 (20)	2 (11.8)	0.61
In-hospital mortality	3 (30)	2 (11.8)	0.32

Continuous variables are presented as mean ± SD or median [IQR], and categorical variables are presented as absolute frequencies (%); statistical significance is marked with (*) for *p* < 0.05 CPB: cardio-pulmonary bypass; ARDS: acute respiratory distress syndrome; MACE: major adverse cardiac events; VIS: vasoactive-inotropic score.

## Data Availability

The datasets generated and analyzed during the current study are available from the corresponding author on reasonable request.

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
