# Peer review of "Intraoperative Hemoadsorption (Cytosorb™) during Open Thoracoabdominal Aortic Repair: A Pilot Randomized Controlled Trial"

_jcm, 2023, doi:10.3390/jcm12020546_

Round 1
Reviewer 1 Report
This is a laudable initiative in that it studies a practical intervention with potentially direct benefit to patient care. In that sense, the study is valuable. Some methodological choices need clarification so the conclusions can be better supported by the results. Slightly more worryingly, the current pilot study does not offer the practical basis for carrying out a bigger multi centre study - more about that below. A few constructive suggestions are made.
Title, abstract
1. These may require some modifications if the suggestions made are accepted. Instead of reporting mean CPB time (without giving the values in both groups) it would be better to say that the majority of baseline and perioperative characteristics were similar.
Introduction and premise
2. The fact that there is a RCT in this area is remarkable. The authors would do well to mention ref [31] from the outset.
3. Another thing that needs clarifying is that the intervention is only intraoperative. 27 patients in this study have spent on average close to 600 hrs on a ventilator and 5 have died. Of those developing serious complications, I presume many took place well beyond the first 48 hrs. And that ongoing SIRS of various intensities was a feature of many cases. Without using the intervention in ICU (for example in those needing dialysis), judging complications at such a distance from the intervention is perhaps less accurate. This does not necessarily need to be expanded on here but it sets the scene for methods and results.
Methods
This is a section that needs quite a bit of work.
4. The STROBRE checklist applies to observational studies. For trials the authors need to use the CONSORT list and provide it to reviewers and readers, even as a supplement.
5. Primary and secondary endpoints - this terminology is very apposite for RCTs but it is not meaningful without a power calculation to determine the numbers needed to enrol. This is missing in the study and I suppose for this reason it is called a 'pilot'.
6. Blinding. This is a major feature in RCTs, harder to do in the operating room - but was it attempted in ICU?
7. Were the patients consecutive? Including 3 non-randomised patients may need a comment from a statistician. I doubt that it has made a significant difference alongside other shortcomings of the design.
8. Cytokine measurement. The lack of it is appropriately quoted as a limitation. If this had been included in at least a few patients it would have proved that the biological basis for the treatment works as intended.
9. VIS. Using this as an endpoint makes a lot of sense, provided that the maximum value is taken in the first 48 hrs or so (that is, close to the operation). It I conceivable that a patient who dies one month down the line in ICU will also have a high VIS prior to their demise, and this must have a very weak connection with the intraoperative intervention.
10. Def of clinical endpoints. These have to be clarified as a lot of the paper is based around interpreting clinical findings. The authors report that all patients who have pulmonary problems have ARDS? Patients ventilated for a long time can have atelectasis, pleural effusions and pneumonias too. How was renal failure defined? By the need for dialysis or using the KDIGO classification?
Results
11. Shorter operation. This does not need to be mentioned as if the intervention was in any way responsible for it. It may be better to say that most perioperative factors were comparable and then focus on those that weren't.
12. Ventilation time. According to Table 2 the difference between the groups is 115 min whereas the text repeatedly says 144 min - which is it?
13. Renal failure. The number of patients dialysed in Table 2 does not reflect the percentages of this complication given in the text (hence I was asking about the definition of renal failure as an endpoint here).
Discussion
14. Previous work. More emphasis needs to be placed on the fact that a previous randomised study was not positive in terms of the intervention.
15. Future work. If other authors wanted to take this further (for example in a multi centre study, as suggested), what would this be powered on? The authors should give us a better numerical or graphic description of the primary endpoint VIS. Or another endpoint, if they prefer. Only then others can make a calculation around the numbers needed to enrol to answer this question definitively.
Reviewer 2 Report
- In Page 2 line 69-73, for randomization, authors have used the sealed envelope method. Why was the ratio of patients not close to 1:1?
- For surgical procedure, did authors use mild hypothermia or normothermia during partial bypass? According to Table 2, it seems authors used mild hypothermia in all cases. It should be stated in the method part. Also, intraoperative respiratory management needs to be mentioned since Control group had such a higher rate of ARDS. For example, when you start one-lung ventilation, recruitment method, etc.
- In this report, ventilation time was quite long (502 hrs = approximately 21 days in HA- group, and 617 hours = 25 days in Control group). What if you used median instead of mean? By the way, this result is totally different from my practice so that I have difficulty with understanding. If these results are true, intervention of HA did not make a difference in terms of meaningful improvement as both intubation durations are extremely long. In line with this, how many patients did fall into “prolonged ventilation” category, and how many patients did require tracheostomy?
- It is also hard to believe this outcome because there was a difference or trend towards the difference in hard endpoints while there was not difference in soft endpoints (inflammatory marker, vasoactive-inotropic score, etc)
- How about blood products use, any difference?
